# Biological Assay-Guided Fractionation and Mass Spectrometry-Based Metabolite Profiling of *Annona muricata* L. Cytotoxic Compounds against Lung Cancer A549 Cell Line

**DOI:** 10.3390/plants11182380

**Published:** 2022-09-12

**Authors:** Edcyl Lee O. Salac, Michael Russelle Alvarez, Rnie Shayne Gaurana, Sheryl Joyce B. Grijaldo, Luster Mae Serrano, Florence de Juan, Rowell Abogado, Isagani Padolina Jr., Froila Marie Deniega, Kimberly Delica, Kimberly Fernandez, Carlito B. Lebrilla, Marlon N. Manalo, Francisco M. Heralde III, Gladys Cherisse J. Completo, Ruel C. Nacario

**Affiliations:** 1College of Arts and Sciences, University of the Philippines Visayas, Iloilo 5023, Philippines; 2Institute of Chemistry, University of the Philippines Los Baños, Laguna 4031, Philippines; 3Department of Chemistry, University of California, Davis, CA 95616, USA; 4Core Lab, Pascual Pharma Corp, Laguna 4030, Philippines; 5Lung Center of the Philippines, Quezon City 1100, Philippines

**Keywords:** *Annona muricata*, lung cancer, metabolomics, cytotoxicity

## Abstract

*Annona muricata* L. (Guyabano) leaves are reported to exhibit anticancer activity against cancer cells. In this study, the ethyl acetate extract from guyabano leaves was purified through column chromatography, and the cytotoxic effects of the semi-purified fractions were evaluated against A549 lung cancer cells using in vitro MTS cytotoxicity and scratch/wound healing assays. Fractions F15-16C and F15-16D exhibited the highest anticancer activity in the MTS assay, with % cytotoxicity values of 99.6% and 99.4%, respectively. The bioactivity of the fractions was also consistent with the results of the scratch/wound healing assay. Moreover, untargeted metabolomics was employed on the semi-purified fractions to determine the putative compounds responsible for the bioactivity. The active fractions were processed using LC-MS/MS analysis with the integration of the following metabolomic tools: MS-DIAL (for data processing), MetaboAnalyst (for data analysis), GNPS (for metabolite annotation), and Cytoscape (for network visualization). Results revealed that the putative compounds with a significant difference between active and inactive fractions in PCA and OPLS-DA models were pheophorbide A and diphenylcyclopropenone.

## 1. Introduction

Lung cancer is by far the most common cause of cancer deaths in both men and women worldwide, accounting for 1.80 million deaths in 2020, or approximately 18% of all cancer deaths [1]. It is categorized into two main types based on prognosis and treatment: non-small-cell lung cancer (NSCLC), which accounts for around 85% of cases, and small-cell lung cancer (SCLC), which accounts for the remaining 15% [2].

Current treatment options for lung cancer include surgery, radiotherapy, chemotherapy, targeted therapy, and/or immunotherapy, depending on the type and stage of cancer. Despite recent breakthroughs in cancer treatment, the prognosis for lung cancer patients remains unsatisfactory. Radiotherapy and chemotherapy have severe side effects due to their cytotoxicity to normal cells, whereas targeted therapy and immunotherapy have a limited target range and can be very expensive [3]. In an attempt to overcome the limitations and drawbacks of these individual and combined cancer treatments, several anticancer drugs have been developed. The effectiveness of most of these drugs is compromised by the ability of cancer cells to acquire drug resistance. Thus, continued research into developing novel and unique anticancer drugs that are more potent, tumor-selective, and can overcome drug-resistant cancer cells is urgently needed [4].

Natural products, particularly those derived from plants, have long played an important role in drug discovery. Medicinal plants with a long history of ethnopharmacological use have been a valuable source of effective phytochemicals that provide beneficial effects against numerous diseases [5]. Approximately 25% of clinically used drugs are derived from plants, with more than 60% of these drugs having anticancer activity. Medicinal plants are often utilized in the form of concoctions or concentrated extracts. Modern medicine, however, requires the separation and purification of one or two active compounds that elicit the activity [6]. Extracts with potential bioactivity are typically subjected to bioassay-guided isolation, which involves (i) extraction of metabolites using appropriate solvents, (ii) chromatographic fractionation of the resulting extract, (iii) bioassay screening of each fraction, (iv) isolation of the molecule(s) from bioactive fractions, and (v) identification of the isolated molecules and evaluation of their bioactivity [7].

One medicinal plant with a long history of ethnomedical use in cancer treatment is *Annona muricata* L., also known as *guyabano*. This evergreen plant is a member of the Annonaceae family and is widely distributed in the rainforests of Central and South America, Africa, and Southeast Asia. It has alternative names such as soursop, *guanábana*, *guanábano*, *guanavana*, *guanaba*, *corossol épineux*, *huanaba*, *toge*-*banreisi*, *durian benggala*, *nangka blanda*, and *cachiman épineux*, among others. It grows around 5–10 m tall and has a diameter of around 15–83 cm. It is typically found at altitudes below 1200 m above sea level, with temperatures ranging from 25 to 28 °C, relative humidity ranging from 60% to 80%, and annual rainfall above 1500 mm [8]. Studies show that extracts and metabolites from guyabano, particularly the leaves, reduce tumor growth and inhibit the formation of various types of cancer cells [9,10,11]. However, the majority of these studies used crude plant extracts rather than bioactive isolates. Thus, the main objective of this study was to extract and partially purify the bioactive compounds present in *A*. *muricata* leaves through sequential solvent extraction and column chromatography. Specifically, it aimed to determine the bioactivity of the fractions from the ethyl acetate leaf extract and identify putative compounds from the fractions that could be responsible for the bioactivity through MS-based metabolite profiling. 

## 2. Results

### 2.1. Bioassay-Guided Fractionation of A. muricata Ethyl Acetate Leaf Extract

A preliminary study was conducted to determine which of the three crude extracts (n-hexane, ethyl acetate, and methanol) obtained from guyabano leaves is the most active. Among the extracts tested, the ethyl acetate extract had the strongest cytotoxic effect in A549 cancer cells (Figure 1). Following this, the ethyl acetate extract was subjected to column chromatography on silica gel 60 using mixtures of hexane, ethyl acetate, and/or methanol as eluents in order of increasing polarity. In the first purification step, a total of 18 fractions were obtained, with fraction F15 having the best anticancer activity. Successive fractionation of fraction F15 resulted in the separation of the two most active fractions, F15-16C and F15-16D. All fractions were tested for bioactivity against the A549 lung cancer cells using the MTS and scratch assays. A summary of the bioassay screening results of GE fractions is illustrated in Figure 2. 

In the first purification, six fractions showed high bioactivity (% cytotoxicity ≥ 50%) at 10 µg/mL in the MTS assay when compared to positive control docetaxel. Fraction F17 (80.65%) was the most toxic to cancer cells, followed by fractions F16 (79.06%), F15 (69.86%), F18 (54.95%), F13 (54.56%), and F12 (51%). Considering the results of both assays, F15 was selected for further fractionation. In the second purification step, fraction F15-16 exhibited the highest %cytotoxicity, followed by F15-15. Further fractionation and analysis of F15-16 resulted in eight subfractions. Of the eight subfractions, fractions F15-16C and F15-16D were the two most active fractions, exhibiting the highest %cytotoxicity values of 99.6% and 99.4%, respectively. Current efforts are being undertaken to further characterize the bioactivities and metabolites of F15-16C and F15-16D.

### 2.2. Metabolomics Profiling of A. muricata Ethyl Acetate Leaf Extract

The chemical profiles of the metabolites obtained from LC-MS/MS analysis of GE fractions were analyzed using the integration of the following metabolomic tools: MS-DIAL (v.4.60, RIKEN), a spectral deconvolution software program for MS data; MetaboAnalyst (v.5.0, https://www.metaboanalyst.ca/, accessed on 5 September 2022), a web-based software program for comprehensive metabolomics data analysis; GNPS (https://gnps.ucsd.edu/, accessed on 5 September 2022), a web-based library for MS/MS spectra; and Cytoscape (v.3.8.2), a software platform for visualizing molecular interaction networks. Before metabolite identification and statistical analysis, the large MS data obtained were pre-processed using the MS-DIAL. Normalization was applied to the raw data to reduce fluctuations and increase confidence in data comparisons [12]. The normalized MS-DIAL data was saved as a .csv file and uploaded to MetaboAnalyst. Non-targeted principal component analysis (PCA) was used to visualize the clustering and trends of the fractions obtained in the third purification—subfractions of F15-16 (active) and F15-17 (inactive). The more similar the sample data, the closer the points in the PCA score plot [13]. The chemical features or metabolites most likely to be responsible for the bioactivity were then subjected to orthogonal partial least squares discriminate analysis (OPLS-DA) to further verify and confirm the initial conclusion in PCA. PCA and OPLS-DA results are presented using score and loading plots.

In Figure 3a, the first component (PC 1) explains the most variation (28.4%), while the second component (PC 2) shows the second highest amount of variance (19%). The scores plot shows that six samples overlap, indicating that they are likely to be found in both active and inactive fractions. Similarly, the PCA scores plot of F15-17 subfractions in Figure 4a shows an overlapping of components. Although the active and inactive fractions were separated, it is possible that the analytes were co-eluted or that the fractions were not fully separated. An ideal PCA plot is when there is a distinct separation between components. However, it is important to take note that the analysis of metabolomics data is complicated due to the inherent variability in each sample and several other experimental or environmental factors, such as the kind of sample used and how the samples were obtained [14]. According to Worley and Powers (2012), PCA scores only show separation between groups when “within-group variation” is much less than “between-group variation” in the data, while PLS scores may show separation at random [15]. Therefore, there is a higher possibility of obtaining biologically relevant results when PCA scores are guided by PLS classification. As shown in Figure 3b and Figure 4b, the active and inactive fractions of F15-16 and F15-17 were separated so that the inactive components (green) were on the far-right side and the active components (red) were on the far-left side. The horizontal component shows the relationship between the active and inactive fractions, while the vertical component shows the relationship of the individual members within the group. The members in the active and inactive groups overlapped with each other since they were just replicates of each other. The OPLS-DA loading S-plot (Figure 3c and Figure 4b) obtained shows the vital features that contributed to the separation of the components. The m/z and rt values with the most negative values of covariance and correlation significantly contributed to the separation between the active and inactive components. The top 20 features were chosen and are presented in Table 1 and Table 2. In addition, the presence of these top 20 features in the active and inactive fractions had a significant difference, as shown in Appendix A. The compound’s m/z and retention times were used in determining the compounds through MS-DIAL. 

### 2.3. Molecular Networking of Identified Compounds in A. muricata Ethyl Acetate Leaf Extract

Several putative compounds were identified from the feature-based molecular networking (FBMN) method in GNPS and Cytoscape analyses of F15-16 and F15-17 fractions. They are observed to be more significant in the active fractions than the inactive fractions. As shown in Figure 5, these compounds are ID 11257 (and 11258), identified as pheophorbide A, with an m/z and rt of 593.28015/20.868, and ID 789, identified as the benzenoid diphenylcyclopropenone, with an m/z and rt of 207.06885/10.591. Between the two compounds, pheophorbide A has a history of use in cancer treatment as a photosensitizing agent. Pheophorbide A is a chlorophyll catabolite that belongs to the porphyrin class [16,17]. Jonker et al. used pheophorbide A as a fluorescent substrate for ABCG2 (also known as breast cancer resistance protein, BCRP), a member of the ATP-binding cassette family of drug transporters that provides resistance to various anticancer drugs [18]. On the other hand, Robey et al. used pheophorbide A as a specific probe for ABCG2 function and inhibition [19].

Moreover, compounds that are significantly present in the inactive fractions and may not contribute to the bioactivity were ID 5287, with an m/z and rt value of 415.20721/16.312 (a flavonoid austinoneol); ID 2318, with m/z and rt value of 303.04907/7.486 (a flavonoid quercetin); ID 2189, with m/z and rt value of 275.99554/9.465 (an aporphine 8H-Benzo[g]-1,3benzodioxolo[6,5,4-de]quinolin-8-one), and ID 7511, with m/z and rt value of 465.11206/7.534 (a flavonoid-3-O-glycoside isoquercitrin). Other compounds identified but with no significant difference in the active and inactive fractions were ID 1682, with an m/z and rt value of 273.07602/11.566 (a flavanone naringenin); ID 1392, with an m/z and rt value of 255.06786/11.576 (an antioxidant chrysin); ID 1433, with m/z and rt value of 256.26559/20.081 (the xanthine dyphylline); ID 5197, with m/z and rt value of 387.19971/5.265 (a triterpenoid roseoside); ID 3189, with m/z and rt value of 317.06598/12.154 (a flavone isorhamnetin); and ID 5972, with m/z and rt value of 413.1936/22.248 (a pyridopyrimidine 9-OH-risperidone).

A prior study by Raheem et al. highlighted the application of combined metabolomics and bioactivity-guided approaches for the successful isolation of a norlanostane-type saponin glycoside with significant antitrypanosomal activity from British bluebells (*Hyacinthoides non-scripta*) plants [20]. In this paper, the pre-processed LC-MS/MS data were subjected to GNPS for metabolite annotation, and the metabolite’s networks were then visualized using Cytoscape. The bioactive metabolites from the antitrypanosomal active fractions were predicted using the OPLS-DA loading S-plot. In another study, Yang et al. successfully screened the active THR/FXa inhibitors from *Salvia miltiorrhiza* Bunge or Dashen, a well-known traditional Chinese medicine with anticoagulant action, using LC-MS combined with multivariate statistical analysis [13]. PCA and OPLS-DA models were used to identify the four bioactive marker compounds (tanshinone IIA, cryptotanshinone, tanshinone I, and dihydrotanshinone I), which are the main active ingredients of Danshen [21]. 

The bioactive compounds from *A. muricata* leaf fractions identified in the current study were previously studied for anti-proliferative activity and cytotoxicity. Rutin from *A. muricata* leaves was also shown to be an effective antitumor agent in an in vivo model using BALB/c mice induced with PC3-cells (prostrate) and 4T1 cells (breast), with results showing approximately a 62% tumor volume reduction and ED_50_ of 10.8 mg kg^−1^ active extracts, respectively [22]. The synergistic effect of *A. muricata* extracts containing quercetin against cancer was also observed either with doxorubicin (breast) or among other flavonoids and acetogenins (prostate) [23]. 

This is the first study, to the best of our knowledge, to show that the following compounds from a member of the Annonaceae family are putative against cancer: 1,4a-dimethyl-9-oxo-7-propan-2-yl-3,4,10,10a-tetrahydro-2H-phenanthrene-1-carboxylic acid, 2-(3,4-dihydroxyphenyl)-5,7-dihydroxy-3-[(2S,3R,4S,5S,6R)-3,4,5-trihydroxy-6-(hydroxymethyl)oxan-2-yl]oxychromen-4-one, 2,4,7,9-tetramethyl-5-decyne-4,7-diol, 3-[(2S,3R,4S,5S,6R)-4,5-dihydroxy-6-(hydroxymethyl)-3-[(2S,3R,4R,5R,6S)-3,4,5-trihydroxy-6-methyloxan-2-yl]oxyoxan-2-yl]oxy-5,7-dihydroxy-2-(4-hydroxy-3-methoxyphenyl)chromen-4-one, 7-[(2S,3R,4S,5S,6R)-4,5-dihydroxy-3-[(2R,3R,4R,5R,6S)-3,4,5-trihydroxy-6-methyloxan-2-yl]oxy-6-[[(2R,3R,4R,5R,6S)-3,4,5-trihydroxy-6-methyloxan-2-yl]oxymethyl]oxan-2-yl]oxy-6-methoxychromen-2-one, 7-[4,5-dihydroxy-6-(hydroxymethyl)-3-[(2S,3R,4R,5R,6S)-3,4,5-trihydroxy-6-methyloxan-2-yl]oxyoxan-2-yl]oxy-2-(3,4-dihydroxyphenyl)-5-hydroxychromen-4-one, 9-OH-risperidone, 9,12-octadecadiynoic acid, adenosine, austinoneol, chrysin, cis-7,10,13,16-docosatetraenoic acid, dioctyl phthalate, diphenylcyclopropenone, dyphylline, isoquercitrin, isorhamnetin, lauric acid leelamide, monoerucin, naringenin, naringenin chalcone|3-(4-hydroxyphenyl)-1-(2,4,6-trihydroxyphenyl)prop-2-en-1-one, naringenin-7-O-glucoside, octadecanamide, ouabain, oxoglaucine, phenanthraquinone, pheophorbide A, roseoside, and sarmentoside B.

## 3. Discussion

The bioassay-guided isolation method is considered the most effective strategy for screening metabolites, particularly when the active component is unknown. Each fractionation step is guided by a bioassay result systematically, reducing the entire processing time and cost [24]. Furthermore, in vitro cytotoxicity assays are typically accompanied by wound healing experiments to determine the efficiency and optimal dose of the tested agents. One of the commonly used in vitro wound healing experiments is the scratch assay. The scratch assay is used to evaluate wound healing by measuring the distance traveled by migrating cells during the assay. A large %wound size indicates that the test extract effectively prevents cancer cells from healing the wound.

In this study, bioassay-guided purification was employed to identify the putative anticancer compounds in the *guyabano* ethyl acetate leaf extract. Numerous studies on the therapeutic benefits of *guyabano* leaves against various human cancers and disease agents have previously been investigated in both in vitro and preclinical animal models. Different classes of annonaceous acetogenins, alkaloids, flavonoids, and phenolic compounds in guyabano extracts have been shown to induce apoptosis and cytotoxicity in cancer cells [8,25]. Moghadamtousi et al. reported that the ethyl acetate extract from guyabano leaves inhibited the spread of A549 lung cancer cells, resulting in cell cycle arrest and apoptosis via activation of the mitochondrial pathway and the NF-B signaling pathway [26]. Similarly, the ethyl acetate extract showed in vivo chemopreventive effects against azoxymethane-induced colonic aberrant crypt foci (ACF) by significantly lowering the ACF formation in rats [27]. 

Recently, metabolomics has received a lot of attention as a practical approach for analyzing a large number of metabolites present in a sample. Unlike the classical approach, metabolomics offers a more efficient and faster process [6]. It involves data acquisition using hyphenated analytical techniques (e.g., GC-MS, LC-MS, or NMR) and data mining or molecular networking using advanced bioinformatics tools (e.g., MS-DIAL, MetaboAnalyst, GNPS, and Cytoscape) to provide extensive metabolites data that help confirm the relationship between a specific compound and its bioactivity. Molecular networking works by either matching experimental spectra against reference spectra (targeted metabolomics) or by aligning experimental spectra against one another and connecting related molecules based on spectral similarity (untargeted metabolomics) to find candidate metabolites directly from fractionated bioactive extracts [28]. Liquid chromatography coupled to mass spectrometry (LC-MS) is currently the most preferred approach for untargeted metabolomics due to its versatility, high throughput, soft ionization, and broad range of metabolites [12,29]. Several LC-MS systems are available, each with unique features that make it better suited to certain applications than others. Untargeted metabolomics often relies on systems with high resolution and MS^2^ capabilities for metabolite identification and relatively fast scan rates to produce enough data points within the short spans of peak elution. Isolating individual components from medicinal extracts may be counter-productive, since they often work synergistically to elicit therapeutic benefits. Nothias et al. (2020) also emphasized that, despite promising bioassay results in the initial extract, the bioactive compound(s) may not be efficiently isolated during subsequent bioassay-guided purification due to the relatively low amounts of bioactive compound(s) present in the fraction or possible degradation [28]. These problems can be addressed by combining the bioassay-guided method with metabolomics studies to predict the putative compounds responsible for the bioactivity. Because metabolomics studies generate a large amount of data, multivariate data analysis must be used to derive conclusions from the results. 

The bioactive compounds from *A. muricata* leaves were extracted and purified using sequential solvent extraction and column chromatography. F15 from the first purification was further purified due to its promising bioactivity. In the second purification, F15-16 had the highest cytotoxicity activity, hence it was chosen for further purification. In the third purification, fractions C and D of F15-16 showed the highest anticancer activity, with %cytotoxicity values of 99.6%, and 99.4%, respectively. The cytotoxicity results were consistent with the scratch assay results, since the fractions were observed to effectively prevent A549 cancer cells from healing the wound. Moreover, subfractions from F15-16 (active) and F15-17 (inactive) were run in LC-MS/MS, and untargeted metabolomics was employed to determine the putative compounds that probably contributed to the bioactivities. The PCA results showed overlapping components, probably due to the co-elution of the compounds during the purification process. OPLS-DA scores plot showed separation of the active and inactive classes and overlapping of the individual elements within each group. The top 50 features from the OPLS-DA loadings S-plot with the lowest values of correlation and variance were obtained, since they significantly contributed to the separation of the active and inactive groups. Feature-based molecular networking of GNPS gave 28 hits or putative compounds for F15-16 fractions. However, most of the hits did not have a significant difference in the active and inactive fractions except pheophorbide A and dicyclopropenone, which were in significantly greater quantities in the active fractions than in the inactive fractions. Moreover, the putative compounds probably contribute to or were responsible for the bioactivities of the active fractions. On the other hand, austinoneol, quercetin, 8H-benzo[g]-1,3benzodioxolo[6,5,4-de]quinolin-8-one, and isoquercitrin were significantly more observed in the inactive fractions and probably do not contribute to the bioactivity. 

## 4. Materials and Methods

### 4.1. Sample Collection, Preparation, and Extraction

Healthy mature *A. muricata* (guyabano) leaves were collected from UPLB Agri Park, Los Baños, Laguna, Philippines. The plant species was verified by Michelle Alejado (Museum of Natural History, University of the Philippines Los Baños), and a voucher specimen (Accession number #073576) was prepared and deposited at the Museum of Natural History, University of the Philippines Los Banos. The fresh guyabano leaves (3.975 kg) were cleaned, cut into small pieces, and air-dried at room temperature for two weeks or until fully dried. The dried leaves (1135.7 g) were homogenized using a blender, and the resulting powdered leaves were macerated with n-hexane (3 × 1.5 L) three times at room temperature for 24 h, with filtering time intervals of 4 h, 4 h, and 16 h. The n-hexane extract was filtered, and the residues were dried and re-extracted sequentially with ethyl acetate (EtOAc) and methanol (MeOH) using the same procedure. The filtrate in each extract was concentrated to dryness using a rotary evaporator and stored in a refrigerator (−20 °C) until further use. 

### 4.2. Bioassay-Guided Fractionation 

The crude guyabano ethyl acetate (GE) extract with the highest %cytotoxicity in the preliminary MTT assay was chosen for further purification. GE extract was subjected to column chromatography over silica gel 60 (0.063–0.200 mm). The column was eluted using n-hexane with a gradient of ethyl acetate up to 100%, followed by increasing the polarity of the mobile phase with methanol. Before column chromatography, solvent optimization was performed using TLC analysis to determine the optimal solvent system (mobile phase). The fractions eluted from the column with similar TLC profiles were combined to yield 18 fractions (labeled as F1 to F18). Each fraction was subjected to cytotoxicity assays on A549 cancer cells. The fraction that exhibited the best anticancer activity, fraction F15, was subjected to the second purification using mixtures of n-hexane, ethyl acetate, and methanol of increasing polarity. The resulting fractions with similar TLC profiles were combined into 17 main fractions (F15-1 to F15-17) and then assayed. The most active fraction (F15-16) was fractionated further to produce a total of 8 fractions (F15-16A to F15-16H), which were subsequently assayed. The most active fractions were F15-16C and F15-16D. Masses of obtained fractions can be seen in Appendix A.

### 4.3. Cell Culture and Bioassay Screening

A549 lung cancer cells (CCL-185TM) were obtained from the American Type Culture Collection (ATCC, Manassas, VA, USA). No further authentication for the cell line was done. The cells were grown in a 20-mL RPMI 1640 medium (Gibco) supplemented with 10% fetal bovine serum (FBS) and 1% penicillin-streptomycin (Thermo Scientific). The media was changed every other day. All cells were grown in at least three biological replicates and maintained in a humidified atmosphere with 5% CO_2_ at 37 °C.

MTS(5-(3-carboxymethoxyphenyl)-2-(4,5-dimethyl-thiazoly)-3-(4-sulfophenyl)tetrazolium) assay was performed to determine the cytotoxicity effects of GE fractions on the A549 cancer cells. A549 cells were seeded into 96-well plates at a density of 3 × 10^3^ cells/well and allowed to adhere for 24 h. Then, the cells were treated with 10 µg/mL GE fractions, 10 µM docetaxel (positive control), and 1% *v/v* DMSO (negative control) and incubated for 24 h at 37 °C, 5% CO_2_. The MTS reagent (CellTiter 96 AQueous One Solution MTS Reagent—Promega) was added to each well containing the fractions and controls and incubated for another 30 min at 37 °C to allow color stabilization. The absorbance was measured at 490 nm using a UV-Vis plate reader, and the cytotoxicity values were reported as % cytotoxicity. One-way ANOVA statistical tests were performed using GraphPad Prism (v.9.4.0, San Diego, USA). Assays were done in triplicate.

Additionally, the scratch/wound healing assay was conducted to assess the inhibitory effects of GE fractions on the migration and wound healing of A549 cancer cells. A549 cells were seeded into 24-well plates and cultured for 48 h to form a monolayer. Then, the cells in the monolayer were scratched with sterile p200 pipette tips and rinsed with PBS twice to remove the detached cells. After which, the cells were treated with 10 ug/mL GE fractions and 0.1% *v/v* DMSO (negative control) and incubated for 48 h. The scratch/ wound closure was monitored and photographed every 24 h after wounding. ImageJ software was used to compute the % wound size. Assays were done in triplicate.

### 4.4. LC-MS/MS Analysis

The bioactive GE fractions were subjected to LC-MS/MS analysis using the Waters Xevo G2-S QTof instrument. The instrument was calibrated and operated according to the manufacturer’s instructions. Dried GE fractions were dissolved in HPLC-grade MeOH at a final concentration of 1000 ppm and eluted through a C-18 column (ACQUITY UPLC, 100 mm, id 2.1 mm, particle size 1.8 µm, particle shape spherical, pore size 100 Å) together with the solvent blank (methanol), QC (quercetin), and standard (mixture of quercetin, gallic acid, naringenin, and rutin). Gradient elution was performed at a 0.4 mL/min flow rate using a mobile phase consisting of 0.1% formic acid in HPLC-grade water (solvent A) and acetonitrile (solvent B). The solvent gradient was as follows: 0 min (95% A; 5% B), 5 min (85% A; 15% B), 15 min (50% A; 50% B), 17 min (30% A; 70% B), 20–22 min (5% A; 95% B), 23–25 min (95% A; 5% B). MS detection was performed using the Electron Spray Ionization (ESI) probe in positive mode. The following ionization parameters were used: sampling cone 75, source offset 80, source temperature 80 °C, desolvation temperature 33 °C. The mass range was selected from 100 to 1000 m/z using 40 V cone voltage, 6 V collision energy, and 30–50 V ramp collision energy. The total run time was 25 min.

The raw data acquired from the LC-MS/MS analysis of GE fractions were processed to identify the putative compound(s) responsible for the bioactivity [30]. Multiple databases such as MS-DIAL [31], MetaboAnalyst [32], FBMN-GNPS [28], and Cytoscape [33] were used for the metabolomics analysis.

The RAW files from the LC-MS/MS run were first converted into analysis base file (ABF) format using the ABF converter before running them in MS-DIAL (v.4.60). In starting a new project in MS-DIAL, soft ionization was chosen, and chromatography for the separation type. For the MS method type, SWATH-MS or the conventional All-ions method was chosen. Centroid and profile data were selected for the MS1 and MS/MS data types, respectively. 

Mass features in the form of [M+H]+ adducts were collected using tolerance values of 0.01 and 0.025 for MS1 and MS2, respectively, with peak heights of 1000 and mass slice widths of 0.1 Da. A sigma window value of 0.5 was employed in the MS2 deconvolution parameters. The MSP file was MS/MS-Public-Pos-VS15, and the MS/MS identification settings had a 100-min retention time tolerance. MS1 had an accurate mass tolerance of 0.01 Da, while MS2 had a tolerance of 0.05 Da, and the identification score cutoff was set at 80%. The alignment result was then normalized and transformed into a comma-separated value (.csv) format to be uploaded in MetaboAnalyst, along with the TIC-peak intensities obtained from the MS-DIAL run.

MetaboAnalyst was used for multivariate data analysis such as PCA and OPLS-DA. The data type uploaded was in a peak intensity table, where the samples are in columns. Because the features are less than 5000, no additional filtering or normalizing was performed. However, log transformation and autoscaling were conducted before starting the analysis. After the analysis, the top 50 metabolites with the lowest p[1] and p[corr] values were obtained.

The mascot generic format (mgf) file and the aligned mass feature table that was initially exported from MS-DIAL were subsequently uploaded to GNPS (Global Natural Products Social Molecular Networking) for further analysis [34]. Feature-based molecular networking (FBMN) was chosen as the analysis method in the GNPS workflow using the following default values: precursor ion mass tolerance: 0.01 Da; fragment ion mass tolerance: 0.05 Da. The network produced was further annotated using three GNPS in silico tools: (1) Dereplicator+ [35], which annotates both peptidic and non-peptidic natural products; (2) MS2LDA [36], which annotates molecular fragments into Mass2Motifs obtained from experimental data; and (3) Network Annotation Propagation (NAP) [37], which propagates the database’s annotations by matching the network further to improve in silico fragmentation candidate structure annotation.

The results obtained from the GNPS run were collated into a single network using MolNetEnhancer [38] and then uploaded to Cytoscape database for molecular networking visualization. The compound taxonomy annotations made by MolNetEnhancer for each compound in the network were noted and cross-referenced to the alignment file generated by MS-DIAL. The MS-DIAL alignment file contains the TIC-normalized peak intensities for each compound in the network, which allowed the determination of the relative abundances of each compound superclass, class, and subclass across sample preparations.

## Figures and Tables

**Figure 1 plants-11-02380-f001:**
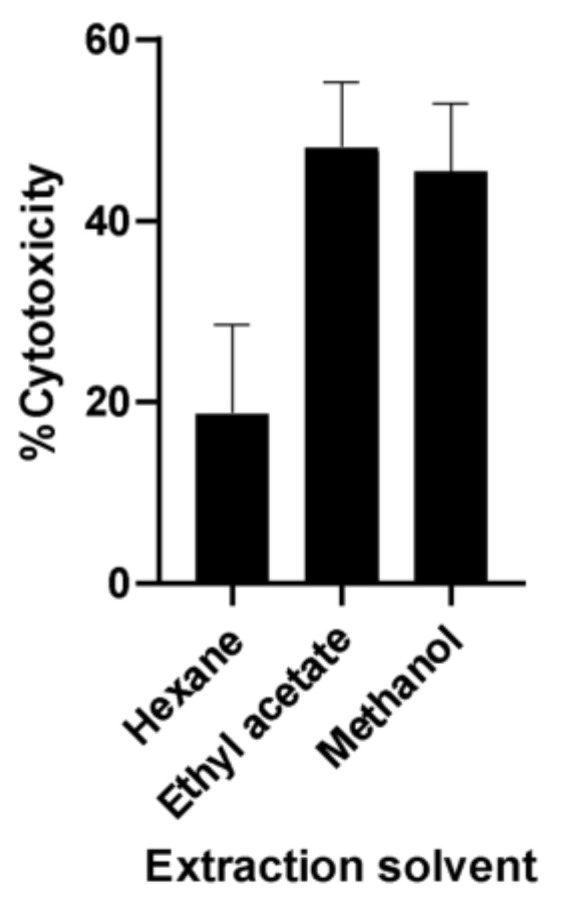
MTT assay screening of crude extracts from *A. muricata* leaves extracted using hexane, ethyl acetate, and methanol.

**Figure 2 plants-11-02380-f002:**
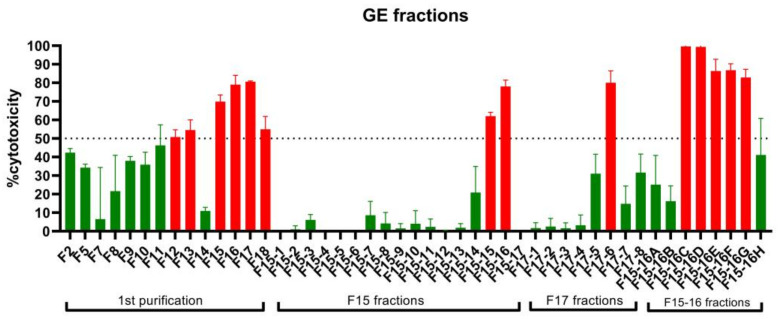
MTS assay screening of partially purified *A. muricata* ethyl acetate fractions after the first, second, and third purification steps. Fractions that are exhibited 50% cytotoxicity are colored red, while fractions below that are colored green.

**Figure 3 plants-11-02380-f003:**
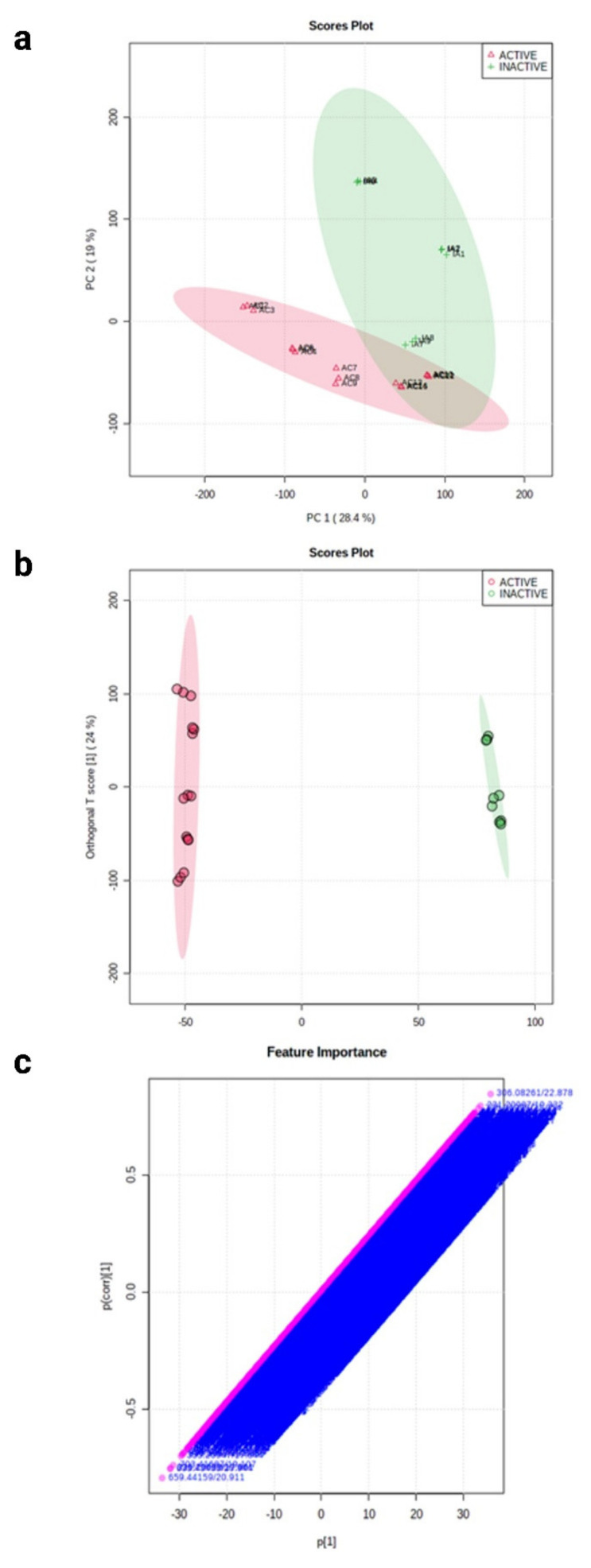
Comparative metabolomics of F15-16 cytotoxic active and inactive fractions. (**a**) PCA scores plot for F15-16 subfractions. (**b**) OPLS-DA scores plot of all of the metabolite features of F15-16 subfractions. (**c**) PLS-DA loadings S-plot for F15-16 subfractions showing the variable importance in a model, combining the covariance and the correlation (p(corr)) loading profile.

**Figure 4 plants-11-02380-f004:**
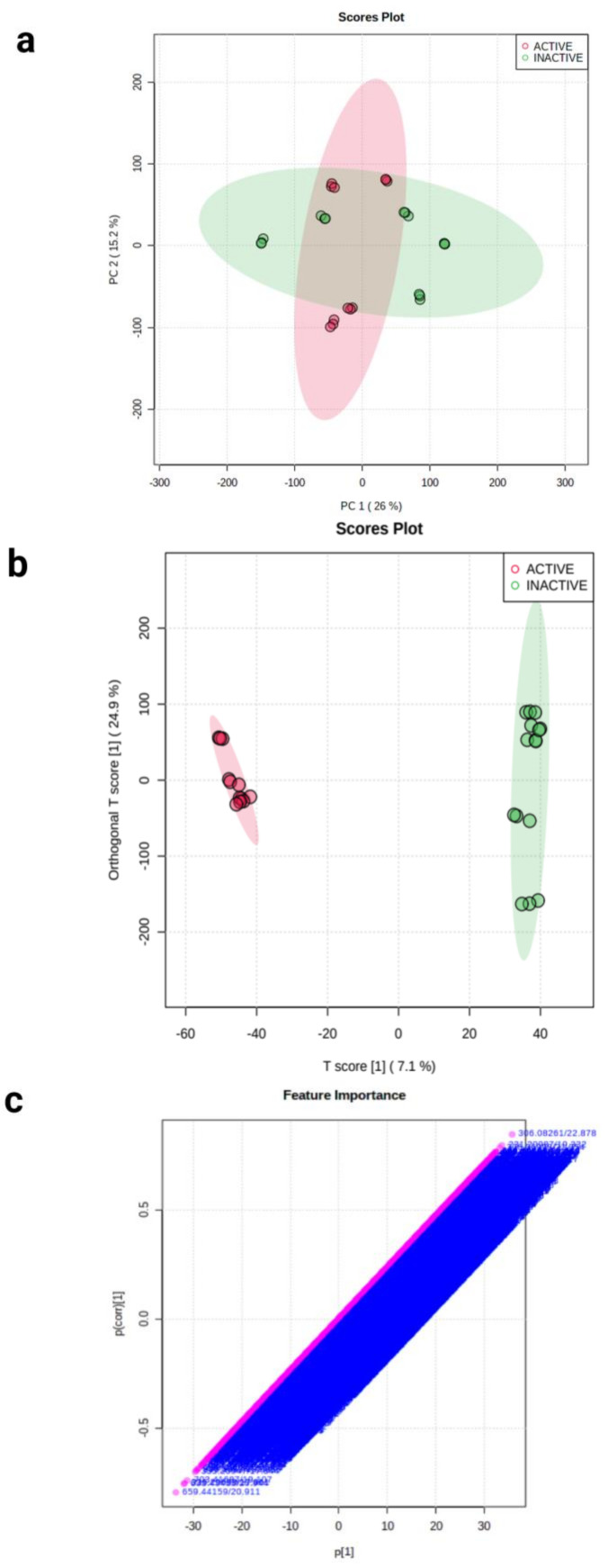
Comparative metabolomics of F15-17 cytotoxic active and inactive fractions. (**a**) PCA scores plot for F15-17 subfractions. (**b**) OPLS-DA scores plot of all of the metabolite features of F15-17 subfractions. (**c**) PLS-DA loadings S-plot for F15-17 subfractions showing the variable importance in a model, combining the covariance and the correlation (p(corr)) loading profile.

**Figure 5 plants-11-02380-f005:**
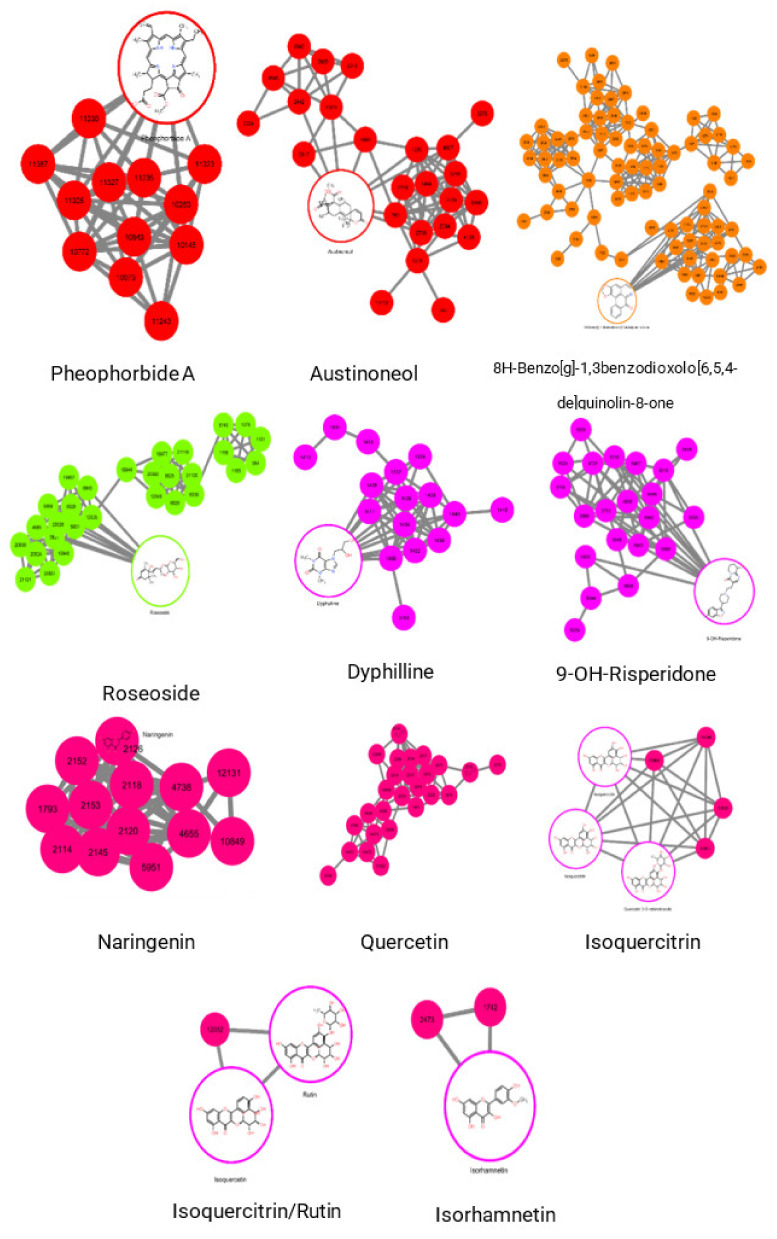
Identified putative compounds and their networks in the F15-16 and F15-17 fractions from Cytoscape analysis.

**Table 1 plants-11-02380-t001:** Suggested compounds in F15-16 subfractions from MetaboAnalyst and MS-DIAL.

Alignment ID	m/z	rt	Formula	Ontology
13271	633.43335	19.473	NA	NA
12388	617.43817	20.733	NA	NA
12357	617.17627	20.72	NA	NA
12376	617.30951	20.734	NA	NA
3078	335.17929	19.474	C21H28O2	Androgens and derivatives
13297	633.61481	19.472	NA	NA
12369	617.27972	20.737	NA	NA
11145	591.42114	18.254	C40H56O2	Xanthophylls
12433	618.26306	20.723	C22H47N5O21S2	4,6-disubstituted 2-deoxystreptamines
12422	617.61731	20.752	NA	NA
11920	609.11182	20.41	C28H23BCl2F4N2O4	NA
11996	610.47235	16.595	C35H63NO7	Macrolides and analogues
11931	609.26654	20.402	C32H42O10	Limonoids
12363	617.23511	20.714	C31H35FN4O7	Dipeptides
13254	633.33405	19.48	C31H50N2O10	Peptides
10417	573.40094	18.334	NA	NA
13144	631.40973	19.491	NA	NA
11923	609.13605	20.392	NA	NA
9844	557.41901	19.811	NA	NA
11985	610.30011	18.351	C35H39N5O5	Ergotamines, dihydroergotamines, and derivatives

**Table 2 plants-11-02380-t002:** Suggested compounds in F15-17 subfractions from MetaboAnalyst and MS-DIAL.

Alignment ID	m/z	rt	Formula	Ontology
15623	659.44159	20.911	NA	NA
14263	633.43433	20.944	NA	NA
3653	336.20688	17.861	C21H25N3O	4-benzylpiperidines
17871	703.41687	19.107	NA	NA
3618	335.20947	17.844	C18H32O4	NA
14942	645.50934	22.086	NA	NA
3722	338.1687	18.385	NA	NA
1551	239.23094	17.983	NA	NA
19912	772.49536	20.734	NA	NA
14387	635.42743	20.948	NA	NA
14941	645.50061	22.144	NA	NA
17456	692.46521	20.672	NA	NA
5453	395.37494	20.871	NA	NA
19882	771.52161	21.33	NA	NA
10807	567.42865	22.217	NA	NA
21854	871.30621	20.854	NA	NA
8693	505.40161	19.697	NA	NA
6588	431.19675	20.974	C22H32O7	NA
3683	337.1861	18.402	NA	NA
17677	699.5072	22.895	NA	NA

## Data Availability

Data available upon request.

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
