# Peer review of "Biological Assay-Guided Fractionation and Mass Spectrometry-Based Metabolite Profiling of *Annona muricata* L. Cytotoxic Compounds against Lung Cancer A549 Cell Line"

_plants, 2022, doi:10.3390/plants11182380_

Round 1

Reviewer 1 Report

The authors studied the bioactive compounds present in Annona muricata leaves against A549 through sequential solvent extraction, column chromatography, and identification of  putative compounds from the fractions that could be responsible for the bioactivity through MS-based metabolite profiling. As the study involves only specific cancer cell line I suggest the change of the title „…Lung cancer A549 cell line“

Lines 241-249              not clear

Material and methods: Mass of the obtained fractions?

Author Response

Dear Reviewer 1,

Thank you for the review. Please see attached our responses to your comments and our revised manuscript.

Reviewer 2 Report

Dear authors, in my opinion, this article is very interesting, good work. I only recommend:

Lines 43-44: “Therefore, the effectiveness of most of these drugs is compromised by the ability of cancer cells to acquire drug resistance”

Lines 137, 203 and so on: Please, change the year to the number of the reference.

Lines 241-249: It is to remove, isn’t it?

Line 340: “% cytotoxicity”

Use hours or h, uniformize along with the article, please.

Line 386: Remove GC

I also recommend two things:

-          Use DAD to quantify the phenolics

-          Perform similar assays with lung non-cancer cells to ensure that the tested concentration is non-toxic to them.

If possible, include a PCA to clarify the obtained results.

Line 52: “Figures 1-2”

Line 175: “more significant”

Line 22: “was also observed”

Line 319: “contribute to or”

What was the criterium to collect the samples?

Revise all English.

Author Response

Dear Reviewer 2,

Thank you very much for the comments. We have revised the manuscript accordingly and have attached here our responses to the comments.

Round 2

Reviewer 2 Report

Dear Authors,

For me, the article can be published.